# Postharvest Immersion in Slightly Acidic Electrolyzed Water Improves Guava Storability by Regulating Phenylpropane Metabolism

**DOI:** 10.3390/foods13233850

**Published:** 2024-11-28

**Authors:** Hongbin Chen, Shujuan Feng, Yazhen Chen, Xuanjing Jiang, Yuzhao Lin, Yihui Chen

**Affiliations:** 1College of Oceanology and Food Science, Quanzhou Normal University, Quanzhou 362000, China; cyz3368@163.com (Y.C.); xuanlikemilk@126.com (X.J.); yuzhaolin820@163.com (Y.L.); 2Key Laboratory of Inshore Resources Biotechnology, Quanzhou Normal University, Quanzhou 362000, China; 3College of Food Science, Fujian Agriculture and Forestry University, Fuzhou 350002, China; shujuan__feng@163.com

**Keywords:** slightly acidic electrolyzed water, guava, quality properties, phenylpropane metabolism

## Abstract

Postharvest guava fruit are at high risk of decay and spoilage, which extremely restrains the further advancement of guava industry in China. Currently, slightly acidic electrolyzed water (SAEW) has been shown to be potent in maintaining the storability of fruits and vegetables. Accordingly, this work was designed to figure out the effectiveness of SAEW on storability maintenance in postharvest guavas via regulating the phenylpropane metabolism. On the harvest day, fresh guavas were immersed in distilled water or SAEW (available chlorine concentration: 30 mg L^−1^) for 10 min, followed by storage for 15 d (25 °C, 80% RH). Results showed that, in comparation with the control guavas, SAEW-treated guavas exhibited lower levels of fruit disease index, malondialdehyde, and cell membrane permeability, while showing higher levels of fruit firmness and commercially acceptable fruit rate, as evidenced by enhanced contents of titratable acid, total soluble solids, vitamin C, total soluble sugar, and reducing sugar. Moreover, SAEW treatment improved the activities of disease-resistance enzymes and the contents of sinapic acid, *p*-coumaric acid, ferulic acid, caffeic acid, and lignin. The above data revealed that SAEW treatment-enhanced storability of guavas was attributed to the increased disease-resistance enzyme activities and disease-resistance substance contents, which improved the fruit disease resistance and slowed down the disease occurrence.

## 1. Introduction

Guava (*Psidium guajava* L.), a fruit tree of the Myrtaceae family, is originally from the tropics of the Americas, particularly in Central and South America [1,2,3]. Nowadays, it is widely distributed in the tropical and subtropical zones around the world, including China, India, and Thailand [2,3]. In China, guava is widespread in Fujian, Hainan, Guangdong, Guangxi, and other southern regions [4]. As a best-selling fresh fruit, the popularity of guava is tied to its high nutritional and medicinal value. Research has confirmed that guava is a rich source of vitamins, pectin, caffeine, chlorogenic acid, and so on, which endow the fruit with anti-inflammatory and hemostatic properties [2,4,5]. Additionally, guavas exhibit antioxidant, hypoglycemic, and hypolipidemic effects due to functional components, such as flavonoids, phenols, and polysaccharides [1].

Nevertheless, as a typical climacteric fruit, guava is highly susceptible to softening and yellowing during postharvest storage due to its high metabolic rate [6,7]. Moreover, these physiological changes augment the affectability of guava fruit to pathogenic fungal infections, accelerating fruit decay and spoilage, which greatly reduces its commercial value and constrains the marketability of postharvest guavas [1,2]. Therefore, it is essential to conduct research on technologies for delaying the disease occurrence and enhancing the storability in guavas.

Slightly acidic electrolyzed water (SAEW) is an aqueous solution with a pH range of 5.0–6.5, an available chlorine concentration (ACC) of 10–80 mg L^−1^, and an oxidation-reduction potential (ORP) over 600 mV. SAEW is created via electrolyzing low concentrations of sodium chloride or dilute hydrochloric acid under direct current voltage [8,9,10,11]. Hypochlorous acid (HOCl), the main active ingredient in SAEW, empowers it with strong bactericidal properties [12,13]. In recent years, due to its wide range of sterilization, environment and human safety, ease of access, and other advantages, more researchers have applied SAEW to the microbial control of fruits and vegetables (FVs). This application has successfully mitigated rotting and quality deterioration in FVs, thus extending the shelf life [11,12,14]. For example, SAEW treatment on carambola fruit significantly maintained fruit storability by preserving higher levels of peel chlorophyll and carotenoids, while reducing the respiration rate and fruit yellowing index [15]. In another study, Li et al. [13] found that SAEW was capable of improving the nutritional quality of broccoli sprouts by enhancing their antioxidant capacity. Similarly, an effective result for improving postharvest storage properties was observed in Chinese bayberry cotreated with SAEW and ultrasonics [9]. Nevertheless, the application of SAEW on guava fruit has not yet been reported. Based on these findings, SAEW treatment, which is cheaper than biological preservation techniques and safer than chemical preservation methods, can be prospected to slow down the decline in guava quality after harvest and to enhance the fruit’s storability.

Prior research has suggested a close relationship between the postharvest storability of FVs and the phenylpropane metabolism, which not only involves the changes in disease-resistance enzyme activities, like phenylalanine ammonia lyase (PAL), cinnamate-4-hydroxylase (C4H), polyphenol oxidase (PPO), cinnamate-4-hydroxylase (C4H), cinnamate dehydrogenase (CAD), and peroxidase (POD), but also the accumulation of intermediate products, like phenolic acids, and their conversion products, such as lignin [16,17,18]. Additionally, chitinase (CHI) and *β*-1,3-glucanase (GLU) participate in the defense response of FVs against pathogenic fungi, regulating the disease resistance in FVs [17]. Higher levels of abovementioned disease-resistance enzyme activities and substances are allowed to mitigate disease incidence and delay the decline of quality attributes, thus maintaining better storability of FVs. This has been reflected in studies on citrus fruit treated with octanal fumigation [19], pear fruit dipped in caffeic acid [20], muskmelon fruit processed by hot water [21], and AEOW-treated longans [22]. However, there is currently no report on the relationship between SAEW treatment and its effects on the storability of postharvest guava fruit in relation to the phenylpropane metabolism. Accordingly, the primary purpose of this investigation was to ascertain how SAEW treatment affects the storability, nutritional traits, disease-resistance enzyme activities, and disease-resistance substance contents of guava fruit during 0–15 d of storage. This information aims to provide guidance for delaying the occurrence of postharvest diseases and inhibiting the quality decline in guavas.

## 2. Materials and Methods

### 2.1. Guavas and Treatments

In the current study, guava (*Psidium guajava* Lour. cv. Hongbaoshi) fruit, which is one of the main cultivars of guava in Fujian, is taken as the fruit material. Fruit of guava cv. Hongbaoshi of approximately 80% physiological maturity (with a peel color chromaticity *L** value of 65.01, a TA range of 1.21–1.25%, and a TSS range of 8.00–8.20%) were collected from an orchard located in Jinjiang City, Fujian Province, China, and transported to our lab on harvest day. The experimental guavas were selected based on having a similar size, color, and the absence of visible damage and diseases.

A slightly acidic hypochlorite water generator BD-600L (purchased from Shanghai Fu-Qiang-Wang Sanitary Products Co., Ltd., Shanghai, China) was employed to generate SAEW. The ORP measuring instrument A57-B (purchased from JIA-BEI Water Treatment Co., Ltd., Guangzhou, China) and ACC measuring instrument RC-3F (purchased from Kasahara Chemical Instruments Corp., Saitama, Japan) were used to gauge the ORP and ACC in SAEW, respectively.

In the pre-experimental stage, a total of 225 selected healthy guava fruits were randomly grouped into 5 sets (45 fruits in each set). One set of fruit was immersed in distilled water for 10 min to serve as the control group. The remaining four sets were each subjected to a 10 min immersion in SAEW with ACC concentrations of 15, 30, 45, and 60 mg L^−1^, respectively. Subsequently, all samples were air-dried under ambient condition and kept at 80% relative humidity and 25 °C, which were considered as the simulated environmental conditions during the harvest season of guavas. Throughout the storage period, the observation and analysis of guava fruit for disease development were conducted every 3 d. After 15 d of observation, it was found that SAEW (ACC: 30 mg L^−1^) had the best effect on disease mitigation in guava fruit, reflected in the FDI of 0.30, which was lower than 0.32 (45 mg L^−1^), 0.35 (60 mg L^−1^), 0.35 (15 mg L^−1^), and 0.65 (control). Therefore, SAEW (ACC: 30 mg L^−1^) was applied to conduct this study.

In the formal experiment, the selected guava fruit were evenly divided into two groups of 250 fruits each. One group was immersed in distilled water for 10 min, which served as the control. The other group was immersed in SAEW (pH = 2.0, ORP = 1700 mV, ACC = 30 mg L^−1^) for the same duration, referred to as the SAEW-treated group. The distilled water and SAEW used for immersing had not been heated or pre-cooled and were all at room temperature. Following immersion, the guavas were allowed to naturally air dry, wrapped in polyethylene film bags (five fruits per bag), and stored for 15 d (25 ± 1 °C, 80% relative humidity). Random samples were taken from two groups every 3 d, and relevant indicators were measured to explore the possible mechanism of the increased storability of postharvest guava fruit induced by SAEW.

### 2.2. Assay of Fruit Disease Index (FDI) and Commercial Acceptability of Fruit Rate (CAFR)

Fifteen guava fruits were subjected to disease grading from 0 to 5 based on the different fruit surface disease areas every three days. The grading scale was as follows: 0, no surface disease areas of the fruit; 1, surface disease areas of the fruit covered less than 25%; 2, surface disease areas of the fruit covered 25% but less than 50%; 3, surface disease areas of the fruit covered 50% but less than 75%; 4, surface disease areas of the fruit covered 75% but less than 100%; 5, surface disease areas of the fruit covered 100%. The specific calculation of the FDI referred to the research by Feng et al. [23]. Among them, fruit graded 0 without any spot or rot and with a good shape and aroma were used to evaluate the CAFR. The percentage of grade 0 guava fruit in the total guava fruit was regarded as the CAFR.

### 2.3. Measurements of Malondialdehyde (MDA) and Cell Membrane Permeability (CMP)

One gram of flesh was collected from 5 guava fruits for the determination of MDA using the procedure described by Menaka et al. [1], with the unit recorded in mmol g^−1^. Additionally, based on the method described by Chen et al. [4], the CMP was measured using 5 guava fruits, and the result was presented as a percentage.

### 2.4. Measurements of Fruit Firmness, L*, h°, and Nutritious Substances

Guided by the method of Chen et al. [4], 5 guava fruits were randomly selected and punctured at four points equidistant from each other on the equatorial plane of the fruit using a texture meter TA. XT Plus (purchased from Stable Micro System Co., Ltd., Godalming, UK). The average force was recorded when a cylindrical probe (2 mm) penetrated the fruit between 3 and 7 mm at a speed of 2 mm per second. Fruit firmness was expressed in N. Additionally, five different guava fruits were examined for the *L** and *h°* values using the methods supported by Chen et al. [4].

The flesh juice of 5 guava fruits was extracted according to the procedure of Hanani et al. [5] to assay the total soluble solids (TSS), which was presented as %.

The measurement of titratable acidity (TA) content in guava fruit followed the procedures introduced by Huyen et al. [24]. The flesh juice (1 g) extracted from 5 guava fruits was homogenized at a ratio of 1: 5 with distilled water and then filtered. To 5 mL of filtrate, 2 drops of phenolphthalein indicator (1%) were added and then titrated with NaOH (0.1 mol L^−1^) until reaching the pale pink endpoint. TA was converted to citric acid for the calculation and expressed as a percentage.

As described by Ribeiro et al. [2], 1 g of flesh juice was extracted from 5 guava fruits to determine the total soluble sugar amount using the phenol–sulfuric acid method, and 1 g of flesh juice from another 5 fruits was taken to measure the reducing sugar content via the 3,5-dinitrosalicylic acid method. Each result was expressed as %.

According to the method recommended by Chen et al. [4] and Feng et al. [23], 1 g of flesh juice was extracted from 5 randomly selected guava fruits to determine the vitamin C (Vc) content, which was presented in mg kg^−1^.

### 2.5. Determination of Disease-Resistance Enzyme Activities

The extraction of disease-resistance enzyme solutions and the subsequent determination of these enzyme activities in the flesh (1 g) from 5 guava fruits per replicate were conducted by referring to the previous study of Lin et al. [17], Li et al. [25], and Chen et al. [26].

In short, PAL enzyme solution (0.5 mL) was added into 3 mL of phosphate buffer saline (PBS) (pH 5.5, 50 mmol L^−1^) and L-phenylalanine (20 mmol L^−1^, 0.5 mL). After experiencing water bathing at 37 °C for 1 h, HCl (6 mol L^−1^, 0.1 mL) was added into the mixed solution to end the reaction, and then the optical density (OD) at 290 nm was measured.

For the C4H activity, 0.5 mL of the C4H enzyme solution was combined with 2.5 mL of Tris-HCl (pH 8.9, 50 mmol L^−1^). After a 30 min water bath at 25 °C, 0.1 mL of 6 mol L^−1^ HCl solution was added to stop the reaction, and the OD at 340 nm was measured.

Additionally, a 4-CL enzyme solution (0.5 mL) was combined with 0.9 mL of the mixed solution (MgCl_2_: ATP: CoA: *p*-coumaric acid = 3: 1: 1: 1). Following a water bath (40 °C, 10 min), the reaction solution’s absorbance at 333 nm was detected.

CAD enzyme solution (1 mL) was combined with equal amounts of 0.1 mol L^−1^ PBS (pH 6.25), nicotinamide adenine dinucleotide phosphate (2 mmol L^−1^), and trans-cinnamic acid (1 mmol L^−1^). After that, the reaction system was immediately placed in the water (37 °C) for thirty-minutes of insulation, and then measured at 340 nm.

The PPO reaction system included the enzyme solution (1 mL), 4 mL of PBS (pH 5.5, 50 mmol L^−1^), and 1 mL catechol solution (0.1 mol L^−1^). The reaction was carried out at 35 °C. After 10 min, the reaction system was cooled immediately, followed by the addition of 2 mL of trichloroacetic acid (TCA) to end the reaction, and then centrifuged. The OD value of the obtained supernatant was measured at 420 nm.

Regarding the POD activity, 3 mL of 50 mmol L^−1^ PBS with a pH of 5.5, guaiacol solution (50 mmol L^−1^, 1 mL), and H_2_O_2_ (1 mL) were combined with POD enzyme solution (0.5 mL). After reacting for 10 min (35 °C), the reaction was stopped with the addition of 2 mL TCA solution, and the OD was detected at 470 nm.

For the CHI activity, 1 mL of the CHI enzyme solution, 2.5 mL of 0.1 mol L^−1^ citric acid-0.2 mol L^−1^ disodium hydrogen phosphate buffer (pH 4.8), and 0.5 mL of 10 g L^−1^ chitin suspension were blended and incubated in a water bath at 37 °C for 1 h. Potassium tetraborate (0.6 mol L^−1^, 0.2 mL) was then added. The mixture was boiled for 5 min. After rapid cooling, 2 mL of 4-dimethylaminobenzaldehyde was added and allowed to react for 1 h at 37 °C. The OD of the supernatant after centrifugation was measured at 585 nm

GLU enzyme solution (1 mL) was mixed with 0.1 mol L^−1^ citric acid–0.2 mol L^−1^ disodium hydrogen phosphate buffer (pH 4.8, 2.5 mL) and laminarin (0.5 mL). The reaction mixture was adjusted to a constant volume of 10 mL with buffer solution after reacting at 37 °C for 1 h. Subsequently, 2 mL of the reaction mixture was combined with 1.5 mL of 3,5-dinitrosalicylic acid and heated for 5 min in a boiling water bath. The mixture was then promptly cooled to room temperature and diluted to a volume of 25 mL using distilled water. The OD of the solution was measured at a wavelength of 540 nm.

The method introduced by Bradford [27] was applied for the detection of protein content in the enzyme solutions, and U mg^−1^ protein was applied to express the enzyme activities.

### 2.6. Assay of Disease-Resistance Substance Contents

The equatorial region fleshes of 5 guava fruits taken from both the control group and SAEW-treated group were ground with liquid nitrogen and dried in an oven (48 h). Three milligrams of dried samples were weighed to assess the lignin content according to the instructions of a lignin content detection kit (Beijing Solarbio Science & Technology Co., Ltd., China), with the results presented in g kg^−1^.

The determinations of *p*-coumaric acid, caffeic acid, ferulic acid, and sinapic acid contents followed the procedures from Luo et al. [28] with some improvements. One gram of pulverized flesh sample from 5 guava fruits was blended with 70% methanol solution (3 mL) and sonicated at 40 Hz for 30 min. Next, the above solution was subjected to centrifugation (4 °C, 12,000× *g*) for 20 min to obtain the sample solution. After that, the prepared sample solution was evaporated to dryness under nitrogen, and the precipitate was dissolved in 1 mL of mixed solution (methanol–distilled water–glacial acetic acid = 70:30:1). The dissolved solution was filtered through a 0.22 µm microporous filter and analyzed by high-performance liquid chromatography (HPLC). An HPLC-2695 system (Milford, Massachusetts, the United States) was carried out on an Inter Sustain^®^ C18 chromatographic column with a column temperature of 25 °C, a flow rate of 0.8 mL min^−1^, and an injection volume of 15 μL. The mobile phase was a mixture of acetic acid solution and methanol solution with a pH of 2.4. The detection wavelengths were 310 nm for *p*-coumaric acid and 325 nm for sinapic acid, ferulic acid, and caffeic acid. Standard curves were created using corresponding standard substances with concentrations ranging from 0 to 100 μg mL^−1^. The results were presented as g kg^−1^.

### 2.7. Data Analysis

This experiment was designed with complete randomization. Three undifferentiated replicates were set up for the determination of the aforementioned indicators. The obtained results were shown as mean ± standard error. SPSS 22.0 (IBM^®^ SPSS Statistics, Armonk, NY, USA) played a role in assessing significant differences (*, *p* < 0.05 and **, *p* < 0.01) between the control and SAEW treatment on the same storage day.

## 3. Results

### 3.1. FDI and CAFR

Figure 1A reflected the changes in the appearance characteristics of postharvest guava fruit at three-day intervals during storage. As storage time increased, the peel color of the guavas progressively changed from green to yellow. Obvious disease symptoms appeared on day 9, manifested by the appearance of black rotting spots, which gradually worsened with prolonged storage.

The FDI (Figure 1B) of guavas began to rise gradually after 6 d of storage. Comparatively, the rise in FDI in guavas was significantly delayed by SAEW treatment. In detail, the FDI reached 0.12 and 0.28 in the SAEW-treated group on days 12 and 15, respectively, much lower than the 0.23 and 0.57 values observed, respectively, in the control guavas.

Figure 1C exhibited that the CAFR of both groups decreased from 6 to 15 d, with a decrease of 93.33% in the control fruit and 53.33% in the SAEW-treated fruit. Undoubtedly, SAEW-treated guavas consistently maintained a higher CAFR than the control guavas during the period from 6 to 15 d.

### 3.2. MDA and CMP

As shown in Figure 2A, the MDA content in the control fruit followed an overall upward trend, except for a brief decline from day 6 to day 9. In SAEW-treated guavas, the MDA content rose steadily until the end of storage after decreasing in the first three days. During storage, the fruit treated with SAEW maintained a lower MDA amount than the control, and the difference between the two reached 35.10% after 15 d.

As reflected in Figure 2B, all samples experienced a constantly rising trend in the CMP throughout the storage period. Moreover, the CMP in SAEW-treated fruit was sustained at a decreased level compared to in untreated guavas. Notably, on days 12 and 15, the CMP in the SAEW-treated fruit was 12.43% and 16.58% lower, respectively, than in the untreated fruit.

### 3.3. Fruit Firmness, L*, h°, and Nutritional Quality

For guavas stored for 15 d, as shown in Figure 3A, the fruit firmness was 3.53 N on the harvest day. As storage time progressed, fruit firmness gradually decreased. On day 15, the fruit firmness dropped to 1.20 N in the control and to 2.16 N in the SAEW-treated group. Especially on days 9, 12, and 15, the fruit firmness in SAEW-treated guavas was notably greater than that in the control guavas, with increases of 0.59-, 0.59-, and 0.80-fold, respectively.

The *L** value in all guava samples enhanced temporarily during 0–3 d, as illustrated in Figure 3B, and then descended gradually within 3–15 d. Compared to the control, the SAEW treatment alleviated the decline in the *L** value in guavas, which was 2.06% and 1.33% greater than the control fruit on days 12 and 15, respectively.

The *h°* value in all guava samples consistently decreased over the 15-day storage period (Figure 3C). Specifically, the *h°* value in the control group dropped from 112.61 to 81.67 after storage, a 27.48% decrease. The SAEW treatment exhibited a decrease of 11.07%, from 112.61 to 101.39. Except for day 3, the *h°* value was higher in the SAEW-treated guavas compared to the control.

As presented in Figure 3D, the TSS content in all guava samples increased within 0–3-day period, peaked on day 3, and then dropped until the end of storage. Additionally, it was observed that the TSS content in SAEW treatment was greater than that of untreated samples on the same storage day. Especially on days 3 and 6, the TSS content in SAEW treatment was 7.53% and 8.64% higher than in the control group, respectively.

After harvest, the contents of TA (Figure 3E) and total soluble sugar (Figure 3F) in guava fruit consistently decreased, regardless of treatment. However, SAEW treatment slowed the decline rate of TA and total soluble sugar contents so that by day 15, their levels were 0.17 and 0.11 times greater than those in the control guavas, respectively.

Additionally, the reducing sugar content in all guava samples rose within the first 3 d, peaked on day 3, and then dropped from day 3 to day 15 (Figure 3G). Compared with control guavas, the SAEW-treated fruit had 4.03%, 11.48%, 10.00%, and 7.82% higher reducing sugar contents on days 6, 9, 12, and 15, respectively.

The Vc amount in guava fruit continued to decline with prolonged storage time (Figure 3H). Compared to the harvest day, the control fruit showed a 65.08% decline in the Vc content, while the SAEW-treated fruit had only a 58.40% decrease. Furthermore, the SAEW treatment increased the Vc content by 19.24% in comparison with the control guavas after 15 d of storage.

### 3.4. Disease-Resistance Enzyme Activities

As shown in Figure 4A, flesh PAL activity rose during 0–9 d and then decreased from 9 to 15 d in all guava samples. Compared to the control, the SAEW-treated group had relatively higher PAL activity during the entire storage process. Especially on day 3, PAL activity in SAEW treatment was 14.12% greater than that in the control guavas.

Additionally, in control guavas, C4H activity increased from 0–12 d and then decreased from 12 to 15 d (Figure 4B). The SAEW-treated guavas showed a similar pattern in flesh C4H activity as the control group. From 3–15 d, C4H activity in guava flesh was boosted by SAEW treatment. On day 12, C4H activity in SAEW-treated guavas was 1.53 times higher, and on day 15, it was 1.55 times higher than in the control guavas.

As illustrated in Figure 4C, control fruit exhibited a 0.13-fold increase in 4-CL activity on day 12 of storage compared to day 0, followed by a decrease from 12 to 15 d. In the SAEW-treated guavas, flesh 4-CL activity exhibited two increases (days 0–3 and 6–12) and two decreases (days 3–6 and 12–15) during the storage period. Compared to the control, the SAEW-treated guavas maintained higher 4-CL activity throughout the storage period.

During the 15-day storage period, all guava fruit showed a trend of gradually rising and then decreasing CAD activity (Figure 4D). It was observed that CAD activity in SAEW-treated guava flesh remained higher than that in the untreated group. Moreover, notable variances of 29.69%, 185.05%, and 277.63% were recorded between the treated and untreated groups at 9, 12, and 15 d, respectively.

The variation tendencies of PPO (Figure 4E) and POD (Figure 4F) activities in all guava samples during storage were similar, as both rose between days 0–9 and descended from days 9 to 15. Furthermore, these two indicators in SAEW-treated guava flesh exhibited higher levels than those of the control guavas during storage.

As displayed in Figure 4G, the flesh CHI activity of control guavas descended slightly during the first 3 days, increased from 3 to 9 days, and then dropped consistently from days 9 to 15. In contrast, the SAEW-treated fruit exhibited rising flesh CHI activity from the day of harvest until the ninth day of storage, and then declined from 7.62 to 2.78 U mg^−1^ protein from days 9 to 15. Throughout the whole storage period, SAEW-treated guavas maintained higher CHI activity in the guava flesh compared to the control.

Additionally, in Figure 4H, GLU activity in control guava flesh experienced a brief decline from 0 to 3 days of storage, a continuous rise from day 3 to day 9, and a subsequent decline from days 9 to 15. Flesh GLU activity in the SAEW-treated fruit increased from days 0 to 9 and then decreased from days 9 to 15, peaking on day 9. In addition, the fruit treated with SAEW showed higher GLU activity throughout the whole storage period compared to the control guavas.

### 3.5. Disease-Resistance Substance Contents

In the current data, the control guava fruit exhibited a 3.57% increment in the *p*-coumaric acid content within 0–9 d, followed by a descending trend for the remaining storage period (Figure 5A). The trend of SAEW treatment on *p*-coumaric acid content was similar to that of the control from 0 to 12 d of storage, showing an initial increase followed by a decrease. However, from 12 to 15 d, a transient increase was observed. In addition, SAEW treatment demonstrated a higher content of *p*-coumaric acid compared to control guavas during storage.

In Figure 5B–E, contents of caffeic acid, ferulic acid, sinapic acid, and lignin in control guava flesh initially ascended and then decreased during the storage period, peaking on days 12, 9, 12, and 6, respectively. Throughout the entire storage period, SAEW treatment encompassed higher contents of these substances than the control guavas. On day 6, the caffeic acid and ferulic acid contents in SAEW-treated fruit were 1.23 and 0.12 times greater, respectively, than in the control fruit. On day 12, the SAEW-treated fruit exhibited 56.84% and 52.73% higher sinapic acid and lignin contents than the control, respectively.

## 4. Discussion

The decline in the storability of FVs is mainly characterized by the degradation of their appearance and the loss of nutritional properties due to disease or senescence, which in turn leads to a drop in commercial value [24,29,30]. For the postharvest guava fruit, poor storability is reflected in fruit softening, loss of nutritional quality, and occurrence of diseases. In this study, the FDI was applied to estimate disease development in guavas, while MDA and CMP were used to objectively evaluate cell membrane damage and aging density in the fruit.

Figure 1 demonstrated that guavas exhibited disease symptoms on day 6, accompanied by an increasing FDI (Figure 1B), which led to a decreasing CAFR (Figure 1C). Furthermore, Figure 6 revealed that the FDI showed a positive correlation with MDA (Figure 2A) and CMP (Figure 2B) in the control fruit, indicating that the destruction of cell structures and senescence in guava fruit accelerated disease development. Additionally, a negative relationship was observed between the FDI and fruit firmness (Figure 3A), *L** value (Figure 3B), *h°* value (Figure 3C), TSS (Figure 3D), TA (Figure 3E), total soluble sugar (Figure 3F), reducing sugar (Figure 3G), and Vc (Figure 3H) in the control group, revealing that the disease occurrence expedited the softening and nutrient degradation in guava fruit, resulting in a poor storability.

Here, SAEW treatment suppressed the increments of FDI, MDA, and CMP compared to the control and maintained higher levels of fruit firmness, peel color, and nutrient contents, thereby achieving a higher CAFR in guavas during storage. A similar impact of SAEW on storability maintenance was found by Zhang et al. [15], namely that SAEW treatment endowed carambolas with higher levels of firmness, TSS, Vc, and TA, while reducing the weight loss rate, thus maintaining fruit quality and reducing physiological loss. Furthermore, Formiga et al. [31] revealed that guava fruit treated with flaxseed protein-based composite coatings exhibited increased fruit firmness and longer retention of green peel color, which contributed to disease control and a prolonged shelf life [31].

The above data show that SAEW was an effective postharvest treatment to sustain the storability of guava fruit, because it inhibited disease development, slowing down the decline in nutritional quality.

A series of defense mechanisms in postharvest FVs are activated during the storage period to resist pathogenic fungal invasion, thus mitigating the disease occurrence and maintaining the storability [16,20,25,32]. Among them, regulating the changes in disease-resistance-related enzymes, i.e., the disease-resistance substance synthetase in the phenylpropanoid metabolic pathway and pathogenesis-related proteins, as well as the disease-resistance substances synthesized in the metabolism of phenylpropane, is a key defense strategy [17,22,25,33,34]. All of these are closely related to slowing down postharvest diseases and reducing quality deterioration in FVs.

The primary enzymes involved in the phenylpropanoid metabolic pathway are PAL, C4H, 4-CL, CAD, PPO, and POD. These enzymes regulate the synthesis of disease-resistance substances in organisms, enhancing the disease resistance of FVs [18,26,35,36]. PAL is responsible for catalyzing the first stage of the phenylpropanoid pathway, converting phenylalanine into trans-cinnamic acid, which is further hydrolyzed by C4H to engender phenolic acids, such as *p*-coumaric acid and caffeic acid [20,33]. 4-CL, as the link between different branches of the phenylpropanoid metabolic pathway, catalyzes the formation of the corresponding acyl-CoA derivatives from these phenolic acids [22,34,36]. Subsequently, the corresponding acyl-CoA derivatives are catalyzed by CAD, POD, and PPO to generate lignin, which acts on strengthening the cell wall of FVs for the enhancement of disease resistance [18,22,35]. Additionally, CHI and GLU are important pathogenesis-related proteins that contribute to the defense response of FVs against pathogenic fungi by degrading *β*-1,3-glucan and chitin in the cell wall of the pathogenic fungi, directly suppressing their growth, thereby alleviating disease development in FVs [17,22,25,33].

Du et al. [35] proposed that elevated levels of PPO, PAL, POD, and GLU induced by benzothiazole helped navel oranges to maintain better fruit quality during storage, manifesting in higher TSS, TA, and Vc contents. Gao et al. [32] found that phenylactic acid could be an effective treatment for maintaining the quality properties of fresh bananas, as it enhanced the activities of the pathogenesis-related proteins CHI and GLU, thus slowing down color change and inhibiting fruit softening. Wang et al. [36] demonstrated that fresh-cut white mushroom stored in high O_2_/CO_2_ conditions promoted PAL, C4H, CAD, and 4-CL activities, contributing to relieving the wounding stress and extending the shelf life. Similarly, Tang et al. [22] declared that the enhanced disease resistance of fresh longans treated with AEOW was attributed to increased activities of CHI, C4H, GLU, 4-CL, and PAL, thus inhibiting disease development and prolonging storage time. As such, it is of great significance to elevate the activity of disease-resistance enzymes to mitigate postharvest diseases and sustain the storability of FVs.

The disease-resistance enzyme activities in all guava samples showed a tendency that enhanced and subsequently declined when the storage time was extended (Figure 4). It could be seen that the self-defense mechanism of guava fruit was gradually activated over time, increasing the activities of PAL (Figure 4A), C4H (Figure 4B), 4-CL (Figure 4C), CAD (Figure 4D), PPO (Figure 4E), POD (Figure 4F), and pathogenesis-related proteins (Figure 4G-H). These enzymes played an active role in defending against fruit diseases. However, in the later storage process, the continuous advancement of fruit senescence and disease disrupted the structure and function of fruit cells, overwhelming the self-defense system and reducing the guava fruit’s resistance to pathogenic fungi. As a result, the activities of disease-resistance enzymes weakened, exacerbating fruit decay, as indicated by rising FDI (Figure 1B) and declining CAFR (Figure 1C) values.

In comparison with control guavas, SAEW treatment retained higher activities of disease-resistance enzymes within 0–15 d, showing fewer disease symptoms (Figure 1A), lower FDI (Figure 1B), and higher CAFR (Figure 1C) during storage. Thus, it was concluded that SAEW effectively promoted defense capability and inhibited pathogenic fungi by maintaining a higher level of disease-resistance enzyme activities in guava fruit, thus delaying disease exacerbation and improving fruit storability. Similar results have been observed in other fruits, such as winter jujubes treated with methyl salicylate and methyl jasmonate [18], longans treated with propyl gallate [17], and pears fumigated with nitric oxide [37].

The phenylpropanoid metabolic pathway uses phenylalanine as the starting material to produce various intermediates, such as phenolic acids, and terminal products, like lignin, through the catalysis of PAL and other enzymes [22,33]. Among them, phenolic acids, including caffeic acid, sinapic acid, ferulic acid, and *p*-coumaric acid act as precursors of lignin biosynthesis, exerting strong antioxidant effects and facilitating the formation of cell structures in FVs [38,39]. In addition, lignin, as a polymer of phenolic compounds, functions in binding to cell wall polysaccharides and cell wall proteins in FVs, thus thickening the cell wall to form a physical barricade for the prevention of pathogenic fungal invasion [17,25]. Therefore, maintaining a high content of disease-resistance substances is of great significance in promoting disease resistance and stabilizing the quality of FVs.

The contents of disease-resistance substances in the control guava fruit ascended first and then descended during the 0–15 d period (Figure 5). Correlation analysis in Figure 6 showed that *p*-coumaric acid (Figure 5A) in the control group was positively associated with PAL (Figure 4A) during storage, while caffeic acid (Figure 5B), ferulic acid (Figure 5C), and sinapic acid (Figure 5D) were all positively correlated with C4H (Figure 4B) and 4-CL (Figure 4C), indicating that phenolic acids involved in lignin synthesis were regulated by disease-resistance substance synthase. Specifically, in the early storage stage, the generation of disease-resistance substances in guavas was increased by the activated disease-resistance substance synthetase activities. However, during the later storage stage, the synthesis of disease-resistance substances correspondingly declined due to the descending disease-resistance substance synthetase activities. This led to weak disease resistance in guavas, which could be reflected in the increased FDI (Figure 1B) and decreased CAFR (Figure 1C).

In addition, the SAEW treatment resulted in a higher content of disease-resistance substances within 0–15 d when compared to the control group, suggesting that SAEW treatment exerted a function in promoting the formation of disease-resistance substances through sustaining higher levels of disease-resistance substance synthase, thus promoting fruit disease resistance and stabilizing fruit quality.

Therefore, the above findings demonstrated the efficacy of SAEW treatment in slowing disease development and stabilizing the quality of guava fruit owing to increased levels of disease-resistance enzymes and substances. This was supported by Li et al. [25], who claimed that phenylalanine treatment inhibited disease incidence and maintained better quality in pears, which was attributed to increased contents of caffeic acid and lignin induced by higher PAL, 4-CL, POD, PPO, and C4H activities. Similarly, Zhang et al. [11] found that SAEW combined with 1-methylcyclopropene activated the PAL, PPO, and POD activities, thus strengthening the disease resistance of fresh-cut kiwifruit and retaining higher quality and nutritive properties.

## 5. Conclusions

The application of SAEW (ACC: 30 mg L^−1^) has been proven to be an effective method for preserving the nutritional qualities and fruit storability of postharvest guavas by regulating the phenylpropane metabolism. The possible mechanism is elucidated in Figure 7. The enhanced storability of guava fruit was primarily due to SAEW treatment promoting the synthesis of disease-resistance substances by facilitating the enhancement of disease-resistance substance synthase activity, thereby strengthening the cell wall structure and resisting the infection of pathogenic fungi in guavas. Additionally, SAEW increased CHI and GLU activities, directly inhibiting pathogenic fungi infection and lessening the disease risk. Due to these synergistic effects, guava fruit treated with SAEW showed a reduced degree of fruit disease, contributing to the retention of high nutrient contents and the prolongation of storability. However, the molecular mechanism by which SAEW enhances the postharvest storability of guava fruit remains unclear. As such, the effects of SAEW on key genes involved in the phenylpropane metabolism should be further explored, which will serve to provide a more scientific basis and a more comprehensive guidance for the application of SAEW in the production practice of guavas.

## Figures and Tables

**Figure 1 foods-13-03850-f001:**
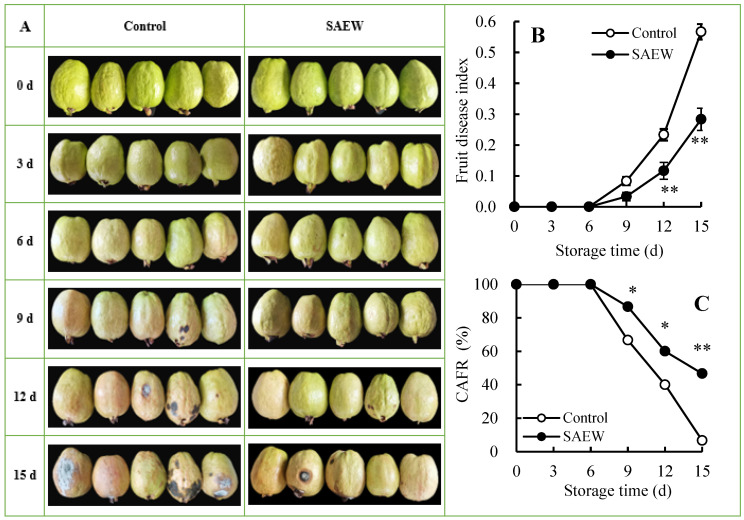
Influences of SAEW on fruit appearance (**A**), FDI (**B**), and CAFR (**C**) in postharvest guavas during 0–15 d of storage. * or ** represent significant (*p* < 0.05) or extremely significant (*p* < 0.01) differences between control guavas and SAEW-treated guavas on the same storage day, respectively.

**Figure 2 foods-13-03850-f002:**
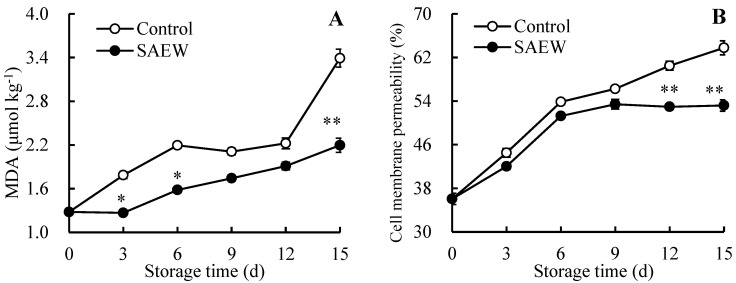
Influences of SAEW on the MDA (**A**) and CMP (**B**) levels in postharvest guavas during 0–15 d of storage. * or ** represent significant (*p* < 0.05) or extremely significant (*p* < 0.01) differences between control guavas and SAEW-treated guavas on the same storage day, respectively.

**Figure 3 foods-13-03850-f003:**
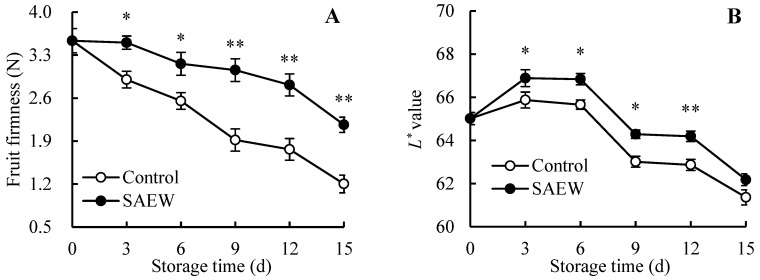
Influence of SAEW on the fruit firmness (**A**), *L** (**B**), *h°* (**C**), TSS (**D**), TA (**E**), total soluble sugar (**F**), reducing sugar (**G**), and Vc (**H**) in postharvest guavas during 0–15 d of storage. * or ** represent significant (*p* < 0.05) or extremely significant (*p* < 0.01) differences between control guavas and SAEW-treated guavas on the same storage day, respectively.

**Figure 4 foods-13-03850-f004:**
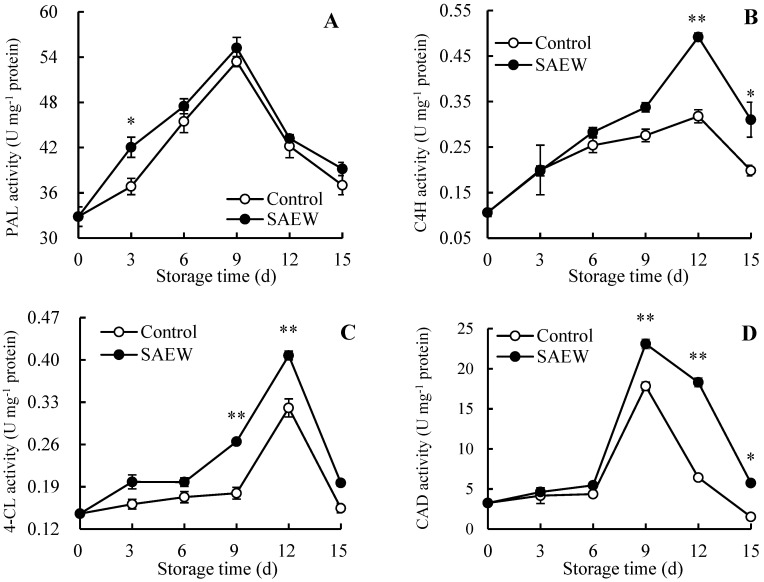
Influence of SAEW on PAL (**A**), C4H (**B**), 4-CL (**C**), CAD (**D**), PPO (**E**), POD (**F**), CHI (**G**), and GLU (**H**) activities in postharvest guavas during 0–15 d of storage. * or ** represent significant (*p* < 0.05) or extremely significant (*p* < 0.01) differences between control guavas and SAEW-treated guavas on same storage day, respectively.

**Figure 5 foods-13-03850-f005:**
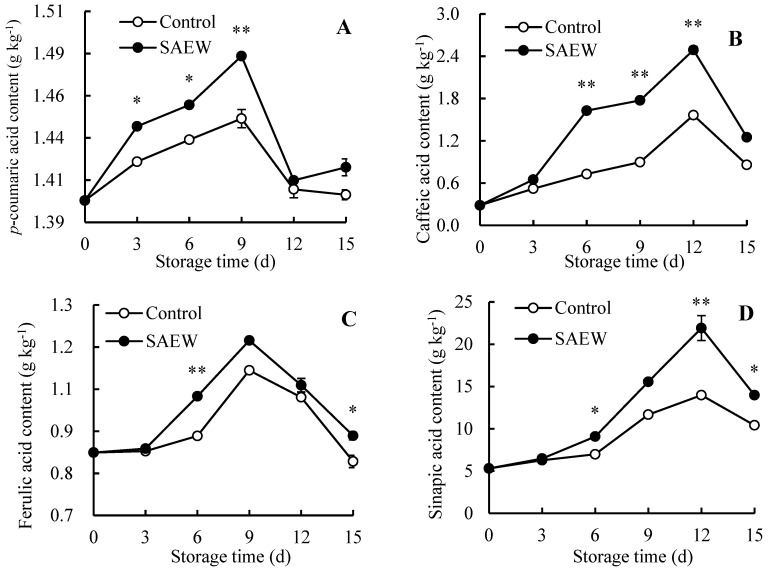
Influences of SAEW treatment on *p*-coumaric acid (**A**), caffeic acid (**B**), ferulic acid (**C**), sinapic acid (**D**), and lignin (**E**) contents in postharvest guavas during 0–15 d of storage. * or ** represent significant (*p* < 0.05) or extremely significant (*p* < 0.01) differences between control guavas and SAEW-treated guavas on the same storage day, respectively.

**Figure 6 foods-13-03850-f006:**
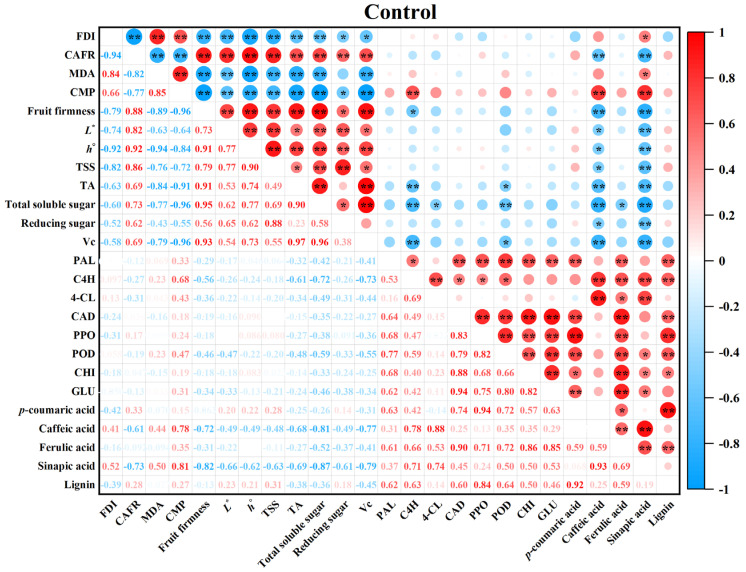
Heatmap of Pearson’s correlations on the obtained indicators in guavas. Positive correlations are presented in red and negative correlations are indicated in blue. The asterisk * or ** in the figure separately denoted the noticeable correlation (*p* < 0.05) or highly noticeable correlation (*p* < 0.01) between these measured indices.

**Figure 7 foods-13-03850-f007:**
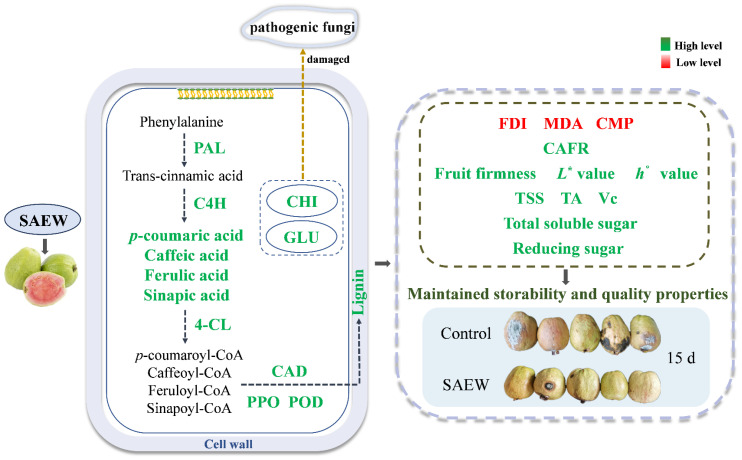
The possible mechanism regarding SAEW treatment enhancing the postharvest storability of guava fruit. The indicators in green indicate a higher level in SAEW treatment when compared to the control, and the indicators in red indicate the opposite.

## Data Availability

The original contributions presented in the study are included in the article, further inquiries can be directed to the corresponding authors.

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
