# Peer review of "Postharvest Immersion in Slightly Acidic Electrolyzed Water Improves Guava Storability by Regulating Phenylpropane Metabolism"

_foods, 2024, doi:10.3390/foods13233850_

Round 1
Reviewer 1 Report
Comments and Suggestions for Authors
The article was well-structured, figures well organized and appropriate for the paper.
The introduction is concise.
The materials and methods align with the research proposal but should be improved.
The results are also well-articulated, were presented very clearly.
The authors provided an excellent discussion of the results, drawing on the work of other researchers.
The conclusion is consistent with the findings.
Despite the overall quality of the article, minor adjustments are necessary to enhance the manuscript.

Author Response
Reviewer #1:
Comment 1:Title should be changed --- “Postharvest immersion in slightly acidic electrolyzed water improves guava storability by regulating phenylpropane metabolism.”
Answer: Thank the reviewer for the suggestion. We have revised it in the new version (See Lines 2-4).
Comment 2:Page 2, Line 60: instead “Another research by Li et al. [11] founded…” should be “Another research, Li et al. [11] founded…”
Answer: Thank the reviewer for the comment. We have revised it in the new version (See Line 61-62).
Comment 3:Page 2, Lines 63-65: instead “Based on these findings, we prospected using SAEW treatment to slow down the decline in the quality of guavas after harvest, and sustain the fruit’s storability.” should be “Based on these findings, the SAEW treatment can be prospected to slow down the decline in guava quality after harvest and to enhance the fruit's storability.”
Answer: Thank the reviewer for the comment. We have rewritten relevant content in the new version (See Lines 66-68).
Comment 4:Page 2, Lines 71-73: "Please merge lines 72-73”
Answer: Thank the reviewer for the comment. We have revised it in the new version (See Lines 74-75).
Comment 5:Page 3, Lines 100-108: The methodology used in the pre-experiment is unclear. The authors mention five sets but only present two. Additionally, the duration of contact with distilled water and SAEW is not specified. Please rewrite this section with more detailed information.
Answer: Thank the reviewer for the comment. According to the revision suggestions, we have added more details about the pre-experiment in the new version (See Lines 104-111).
Comment 6:Page 3, Lines 109-112: What is the temperature of the distilled water and SAEW at the time of immersion? Please clarify whether the process used was submersion, immersion, or soaking, and rewrite the methodology accordingly.
Answer: Thank the reviewer for the comment. We have illustrated the immersion temperature of distilled water and SAEW in the new version, and the methodology has also been rewritten (See Lines 116-122).
Comment 7:Page 4, Line 164: instead “As for the C4H activity”, should be “For the C4H activity”
Answer: Thank the reviewer for the comment. We have revised it in the new version (See Line 173).
Comment 8:Page 5, Line 225: instead “SPSS 22.0”, should be “SPSS 22.0 (IBM® SPSS Statistics, USA)”
Answer: Thank the reviewer for the comment. We have revised it in the new version (See Line 232).
Comment 9:Page 6, Figure 1: I don’t see the need to include (○, Control; ●, SAEW treatment) in the text, as it is already stated in the figure legend.
Page 6, Figure 2: same suggestion.
Page 8, Figure 3: same suggestion.
Page 10, Figure 4: same suggestion.
Page 11-12, Figure 5: same suggestion - Please adjust the size of the graph or modify the text so that the image and legend remain on the same page.
Answer: Thank the reviewer for the suggestions. We have removed the corresponding legend in the figure captions (See Lines 252, 268, 312, 356, and 379).
In order to ensure consistency in the size of the graph, we did not adjust the size of Figure 5. The current layout of this manuscript will be further processed by the editorial department. Before the paper is published, we will communicate with the editor for the final layout, and present the graphs and captions on the same page.
Comment 10:Page 12, Lines 383-384: instead “In our work, we could find in Figure 1 that, guavas exhibited disease symptoms on day 6…”, should be “Figure 1 demonstrated that, guavas exhibited disease symptoms on day 6…”.
Answer: Thank the reviewer for the suggestion. We have revised it in the new version (See Line 389).
Comment 11:Page 13, Line 438: instead “In this study, the disease-resistance enzyme activities…”, should be “The disease-resistance enzyme activities…”.
Answer: Thank the reviewer for the suggestion. We have revised it in the new version (See Line 452).
Comment 12:Page 13, Line 467: instead “In our study, content of disease-resistance substances…”, should be “The content of disease-resistance substances…”.
Answer: Thank the reviewer for the suggestion. We have revised it in the new version (See Line 482).
Comment 13:Page 14, Figure 6: Excellent information, but the authors briefly mentioned Figure 6 at the beginning of the discussion (page 12, lines 385-392). To improve understanding and the final discussion of the results in relation to Figure 6, I suggest placing this discussion before Figure 6 for a more coherent conclusion to the article.
Answer: Thank the reviewer for the suggestion. We have revised it in the new version (See Lines 411-412).
Comment 14:Page 15, Line 493: instead “In the current work, the application of SAEW…”, should be “The application of SAEW…”.
Answer: Thank the reviewer for the suggestion. We have revised it in the new version (See Line 509).
Comment 15:Page 15, Lines 495-496: The authors mention Figure 7 in the conclusion. Although it is quite illustrative, it doesn’t seem appropriate to include it there. I suggest relocating it to the end of the article, right after the discussion, and providing an explanation in the text about the content of the figure.
Answer: Thank the reviewer for the suggestion. Figure 7 is the possible mechanism regarding SAEW treatment enhancing the postharvest storability of guava fruit. However, we think Figure 7 is more appropriate mentioned in the conclusion. Because the conclusion is a highly summarized and concluded part of the entire manuscript. Mentioning Figure 7 in the conclusion can succinctly present the complex mechanism by which SAEW improves the storability of guavas, helping readers better organize and extract the core points of the research. Similarly, in some published papers, the mechanism diagram is also mentioned in the conclusion. Here are some examples:
Wang, L.; Huang, X.L.; Lin, C.C.; Zhang, C.; Shi, K.L.; Wang, M.L.; Wang, Y.Y.; Song, Q.Y.; Chen, X.Y.; Jin, P.; Zheng, Y.H. Hydrogen sulfide alleviates chilling injury by modulating respiration and energy metabolisms in cold-stored peach fruit. Postharvest Biol. Technol. 2023, 199, 112291. doi: 10.1016/j.postharvbio.2023.112291
Lin, Y.F.; Chen, J.; Lin, Y.X.; Lin, M.S.; Wang, H.; Fan, Z.Q.; Lu, W.J.; Chen, Y.H.; Lin, H.T. DNP and ATP modulate the pulp softening and breakdown in fresh longan by acting on the antioxidant system and the metabolisms of membrane lipids and cell wall polysaccharides. Food Chem. 2024, 460, 140531. doi: 10.1016/j.foodchem.2024.140531.

Reviewer 2 Report
Comments and Suggestions for Authors
The manuscript entitled: ”Postharvest SAEW immersion enhances storability of guava fruit by regulating the phenylpropane metabolism” is addressed to study the effect of slightly acidic electrolyzed water on guava fruits to assess the storability maintenance in postharvest and storage of the fruits. Yet interesting for the results reported, the proposed manuscript seems to have possible quite limited overall impact on the overall fuits storage field since no comparison seems to have been considerd with other fruits using the same approach. Some minor and other remarks. The origin of the starting materials (the fruits) should be better detailed. The experimental part, the novelty and the limits of the proposed manuscript should be more exploited and detailed. It is also suggested to avoid the use of abbreviations in the title of the manuscript, and also to define in full any abbreviations at their first use.
Author Response
Reviewer #2:
Comment 1:The manuscript entitled: "Postharvest SAEW immersion enhances storability of guava fruit by regulating the phenylpropane metabolism" is addressed to study the effect of slightly acidic electrolyzed water on guava fruits to assess the storability maintenance in postharvest and storage of the fruits. Yet interesting for the results reported, the proposed manuscript seems to have possible quite limited overall impact on the overall fruits storage field since no comparison seems to have been considered with other fruits using the same approach.
Answer: Thank the reviewer for the comment. The impact of SAEW on the fruits storage field has been mentioned in Lines 55-65. The main objective of this article is to use SAEW as a postharvest treatment to enhance the storability of guava fruit, so our focus is on guavas rather than the overall fruits storage field. According to the suggestion, we still added examples of using SAEW for other fruits in the new version (See Lines 401-404 and 504-507). Applying SAEW treatment to a wider variety of fruits in the future may be an interesting direction for exploration.
Comment 2:Some minor and other remarks. The origin of the starting materials (the fruits) should be better detailed. The experimental part, the novelty and the limits of the proposed manuscript should be more exploited and detailed. It is also suggested to avoid the use of abbreviations in the title of the manuscript, and also to define in full any abbreviations at their first use.
Answer: Thank the reviewer for the comment. We have supplemented the origin of guava fruit in Lines 33-36. The details of experimental part, the novelty and the limits in the manuscript have been refined in Lines 104-123, Lines 65-68, and Lines 519-523, respectively.
Additionally, we have revised the title (See Lines 2-4) and reviewed the article to make sure all abbreviations were defined at their first use.

Reviewer 3 Report
Comments and Suggestions for Authors
Dear Authors,
In general terms, we recommend revising the English language and the wording of the manuscript. Colloquial expressions were observed that do not correspond to a scientific article. It is important to point out to the authors that they are not publishing in a popular journal but in a scientific journal.
The observations in their manuscript are marked in yellow. Some sentences and words need more clarity, vocabulary, sentence consistency, lack of punctuation marks, and correct use of articles or prepositions in different sentences, among other grammatical writing errors.
The need for prepositions, articles, and punctuation marks was observed in several sections. Some of these were pointed out in the manuscript enclosed to this report.
Check if the journal allows using abbreviations in the title because the authors placed SAEW in the title.
Figure 4 of this article is not readable. The values of the graph are not readable.
In my opinion, the manuscript provides very interesting information; I recommend to the authors a better analysis of the data both in the results section and in the discussion. The latter section requires an argumentation with sufficient scientific quality.
The document requires a complete revision of the English language, due to grammatical errors that have been pointed out. The paper should be proofread by a native English speaker. Authors are requested to avoid the use of AI programs for English editing.
Yours sincerely,
The reviewer.

My comment is:
The document requires a complete revision of the English language, due to grammatical errors that have been pointed out. The paper should be proofread by a native English speaker.
Author Response
Reviewer #3:
Comment 1:In general terms, we recommend revising the English language and the wording of the manuscript. Colloquial expressions were observed that do not correspond to a scientific article. It is important to point out to the authors that they are not publishing in a popular journal but in a scientific journal.
The observations in their manuscript are marked in yellow. Some sentences and words need more clarity, vocabulary, sentence consistency, lack of punctuation marks, and correct use of articles or prepositions in different sentences, among other grammatical writing errors.
The need for prepositions, articles, and punctuation marks was observed in several sections. Some of these were pointed out in the manuscript enclosed to this report.
Answer: Thank you for your comment. We have revised the whole manuscript carefully and tried to avoid any grammar or syntax error. In addition, we have invited the native English speaker to review this manuscript, and avoided any mistakes of tense, grammar and syntax in the new version. We believe that the language is now acceptable for publication.
Comment 2:Check if the journal allows using abbreviations in the title because the authors placed SAEW in the title.
Answer: Thank the reviewer for the suggestion. We have revised the title (See Lines 2-4) and reviewed whole manuscript to make sure all abbreviations were defined at their first use.
Comment 3:Figure 4 of this article is not readable. The values of the graph are not readable.
Answer: Thank the reviewer for the comment. In the version we submitted to the editor, the value in Figure 4 was readable. But we still checked all the graphs in this article again to make sure the values of the graphs could be readable.
Comment 4:In my opinion, the manuscript provides very interesting information; I recommend to the authors a better analysis of the data both in the results section and in the discussion. The latter section requires an argumentation with sufficient scientific quality.
Answer: Thank the reviewer for the suggestion. In this article, we expected the result section mainly described the data graph and the in-depth analysis was provided in the discussion section. According to the suggestion, we have added analysis to the discussion section in the new version (See Lines 399-410 and Lines 488-493).
Comment 5:The document requires a complete revision of the English language, due to grammatical errors that have been pointed out. The paper should be proofread by a native English speaker. Authors are requested to avoid the use of AI programs for English editing.
Answer: Thank the reviewer for the suggestion. We have revised the whole manuscript carefully and tried to avoid any grammar or syntax error. In addition, we have invited the native English speaker to review this manuscript, and avoided any mistakes of tense, grammar and syntax in the new version.

Round 2
Reviewer 2 Report
Comments and Suggestions for Authors
The proposed manuscript has been properly revised and the main remarks addressed. It can be further processed.
Reviewer 3 Report
Comments and Suggestions for Authors
Dear Authors,
I have reviewed your manuscript and agree with the modifications made.
I have no observations or comments on the edited manuscript.
Yours sincerely,
The Reviewer